# Program Evaluation and Refinement of the “Safe Functional Home Exercise” Program for Improving Physical Activity in Older People with Dementia Who Receive Home Care

**DOI:** 10.3390/healthcare12020166

**Published:** 2024-01-10

**Authors:** Den-Ching A. Lee, Michele L. Callisaya, Terry P. Haines, Keith D. Hill

**Affiliations:** 1Rehabilitation Ageing and Independent Living (RAIL) Research Centre, Monash University, Peninsula Campus, Frankston, VIC 3199, Australia; keith.hill@monash.edu; 2National Centre for Healthy Ageing, Monash University and Peninsula Health, Frankston, VIC 3199, Australia; michele.callisaya@monash.edu (M.L.C.); terry.haines@monash.edu (T.P.H.); 3Peninsula Clinical School, Monash University, Frankston, VIC 3199, Australia; 4School of Primary and Allied Health Care, Faculty of Medicine, Nursing and Health Sciences, Monash University, Frankston, VIC 3199, Australia

**Keywords:** physical activity, home care, cognitive impairment, exercise, co-design

## Abstract

Introduction: People with dementia who receive home care have low levels of physical activity participation. Objectives: To evaluate and refine a co-designed exercise program for home care clients with dementia, led by trained care support workers. Methods: An action research cycle whereby support workers, clients and carers (each n = 26) from the “Safe Functional Home Exercise” feasibility study were invited to complete an evaluation survey. Participants rated statements using Likert-style scales on (1) staff training, (2) staff confidence, (3) program support and (4) satisfaction. The participants could provide comments for situations that affected exercise performance, suggested improvements for staff training, program support and exercises. The co-design panel (original program designers) was reconvened to refine the exercise program. Results: Support workers (n = 19), clients (n = 15) and carers (n = 13) returned their surveys. Support workers (74–90%), carers (77–92%) and clients (100%) rated program support positively and were overall satisfied. Support workers (>80%) agreed that the training course was relevant and were confident in delivering the exercises to clients. Situations included “Covid isolation”, the client’s “poor medical condition” and “mood fluctuations” that made exercising difficult. Improvements included “making a client video” and “providing tips to motivate clients”. The co-design panel modified the exercise program. Conclusions: The “Safe Functional Home Exercise” program is the first exercise program co-designed for people with dementia. It is well accepted by support workers, people with dementia and carers. Utilising support workers to facilitate physical activity participation is potentially low-cost and scalable in home care. Future studies are needed to evaluate the refined program in home care.

## 1. Introduction

Physical activity is beneficial for older people with dementia [1,2]. Higher physical activity levels reduce the risk of chronic disease and falls and improve balance, mobility, function, psychological health, wellbeing and quality of life in older people [3]. Systematic review evidence supports that various forms of physical activity can be implemented safely and achieve similar outcomes for people with dementia [4,5,6,7]. These programs, including those with balance training, have been safely implemented in people with dementia, often with carer support or in a supervised setting [8]. However, people with dementia typically have a low physical activity level [9], especially among those who receive home care [10]. People who rely on home care to support their living at home are often functionally dependent and homebound and are at an elevated risk of physical inactivity and falls [11]. Low physical activity participation among individuals with dementia or cognitive impairment can lead to dependence on activities of daily living, reduced mobility, a lower quality of life and an expedited decline in cognitive abilities [12,13,14].

The home care program, subsidised by the Australian Government’s aged care system, facilitates older people in receiving assistance to maintain their residence and lifestyle while living at home [15]. Care support workers visit home care clients regularly to provide home care services and are familiar with their clients and family carers. We capitalised on this funding mechanism and the potential for care support workers to facilitate physical activity in home care clients during their care shifts.

Care support workers who receive vocational training deliver home care services such as domestic and personal care; however, there was no focus in their training on how to facilitate safe physical activity participation in home care clients. To facilitate this within the framework of supporting individuals with dementia, it is essential for care support workers to possess the requisite skills, confidence and capability necessary to assist those with dementia in participating in physical activities. Moreover, the advocated physical activities must meet the criteria of being both safe and effective while also aligning with the preferences and acceptance of individuals with dementia. For implementation in home care, it would also require the physical activity program to be of low complexity and low cost with little or no equipment requirement to suit the organisation, care support staff, home care clients and family carers. We co-designed a physical activity program to fit this purpose, called the “Safe Functional Home Exercise” program for people with dementia. We have since completed a feasibility study of this program in which care support workers were trained to deliver the program to home care clients with dementia or cognitive impairment for 12 weeks at their home, once weekly for 15 min during care shifts [16]. This was supplemented by family carers’ supervision of exercises for 30 min, three times weekly for 12 weeks, and fortnightly phone support from a physiotherapist to ensure safety and exercise progression.

The aims of this study were to (1) evaluate the perspectives of care support workers, home care clients and family carers who participated in the feasibility study of the “Safe Functional Home Exercise” program and (2) refine the exercise program based on evaluation findings by reconvening the original co-design panel that developed the program.

## 2. Materials and Methods

### 2.1. Design and Setting

A cross-sectional survey and a co-design meeting were both conducted in a community setting.

### 2.2. Participants

Participants were care support workers, home care clients and family carers who participated in the feasibility study of the “Safe Functional Home Exercise” program, which was conducted in a community setting between April and November 2022 [16]. Care support workers were staff members of two aged care organisations in Australia who attended training and participated in the delivery of the “Safe Functional Home Exercise” program to home care clients, including one organisation providing home care services for a Culturally and Linguistically Diverse (CALD) client group. Home care clients were aged ≥60 years who had any of the following: (a) a dementia diagnosis; (b) a cognitive score of ≤22 on the Rowland Universal Dementia Assessment Scale (RUDAS) [17]; or (c) were receiving a dementia and cognition supplement in their home care packages [18]. Family carers were individuals who had the capacity to support individuals with dementia or cognitive impairment during exercise sessions. Alternatively, they may collaborate with care support workers to collectively fulfil the weekly exercise target for the person in their care.

### 2.3. Survey Instrument

The researchers, who were experienced clinicians in the evaluation of exercise programs for older people, developed a survey for care support workers, home care clients and family carers. This evaluative tool drew inspiration from a survey utilised in an online educational course designed for fitness leaders, specifically focusing on exercise prescription for fall prevention [19]. The surveys were written in the English language and then translated to the Chinese language by the chief researcher (DCAL), who is bilingual and fluent in both languages. All participants rated their level of agreement with three statements regarding program support and program satisfaction. Home care clients and family carers rated these statements on a 3-point Likert scale (ratings of “1” represented agree, “2” represented neither agree nor disagree and “3” represented disagree), while care support workers rated them on a 5-point Likert scale (ratings of “1” represented “strongly agree”, “3” represented “neither agree nor disagree” and “5” represented “strongly disagree”). These statements were:“I feel supported by the research team throughout the exercise program”.“I feel supported by my organisation throughout the program”.“Overall I was satisfied with delivering the exercise program”.

In addition, care support workers rated their level of agreement with seven statements regarding the training course and their confidence in delivering and prescribing the exercise program using a 5-point Likert scale (ratings as above). These statements were:“The program was pitched at an appropriate knowledge level”.“I found the resources are relevant to my learning”.“I feel prepared to prescribe exercise for home care clients with dementia”.“I feel prepared to prescribe exercise for the family carer so that they can supervise the client to exercise in the week”.“The training was relevant for delivering exercise to home care clients with dementia”.“The training was relevant for instructing family carers to supervise home care clients with dementia to exercise”.“Overall, completing the training course has improved my confidence to prescribe exercise for home care clients with dementia and their family”.

Closed-ended questions (with responses Yes/No/Unsure) were used for clients and family carers to assess a range of items on whether they thought (1) doing exercise was important for the client; (2) participating in this exercise program had been beneficial to the client; (3) the care support worker had been adequately trained to deliver this exercise program; and (4) sufficient resources and support were provided to help the client do the exercises each week. In addition to this, closed-ended questions (with responses Yes/No/Unsure) were used for care support workers, clients and family carers to determine whether they thought (1) exercises delivered by care support workers during care shifts should be adopted as a regular practice for home care clients with dementia who were able to participate and (2) if they had encountered any situations that may have positively or negatively influenced participation in the exercise program. Participants could add open-text responses for these situations, their thoughts about adopting the exercise program delivery as a regular home care service, improvements that they felt could enhance program support and the exercise program and suggestions that care support workers felt could improve the training course.

### 2.4. Procedure

This study was approved by the Human Research Ethics Committees of Monash University (ID 28838). A written consent was obtained from all participants and, if required, their person responsible (for people with dementia) prior to participation in the evaluation survey. All participants in the feasibility study (26 care support workers, 26 home care clients and 26 family carers) were invited to complete the program evaluation survey regardless of program completion. The chief investigator (DCAL) provided the participants with the survey link to the electronic Qualtrics survey using WhatsApp or WeChat app or a paper-based survey in person or by post at the end of the 12-week program for program completers or when client/family carer dyads notified the chief researcher of their withdrawal from the feasibility study. The survey was conducted between May and November 2022. The STROBE checklist for cross-sectional studies was followed for reporting this research [20].

As part of a larger study, which had four steps (see Figure 1), this paper describes Step 3 (feedback from care support workers, home care clients and family carers on program participation) and Step 4 (refinement of the physical activity program by providing findings back to the co-design panel) of the action research cycle (Figure 1).

Step 1 (co-design of the physical activity program) and Step 2 (feasibility study of the physical activity program) were completed and described in another paper [16]. Briefly, Steps 1 and 2 involved the following:

Step 1 (Co-design): The “Safe Functional Home Exercise” program for dementia was collaboratively developed by an expert panel comprising exercise and fall prevention researchers, clinical physiotherapists and community aged care stakeholders from Melbourne, Sydney, and Perth (Australia). The co-design process took place in two 2 h meetings held a month apart. In the first meeting, the panel formulated an approach that included a physical activity program and exercise choices, drawing input from panellists, a desktop review by the research team, and tailored implementation principles for home care clients with dementia. The research team synthesised these inputs into an exercise program and staff training resources. In the second meeting, the program underwent a thorough review and refinement, resulting in a comprehensive package addressing safety considerations for care support workers and family carers delivering the exercises. The resulting “Safe Functional Home Exercise” program spanned 12 weeks, featuring four exercises with two difficulty levels each, aiming to improve strength, balance and fitness for people with dementia in home care.

Step 2 (Training and Field Testing): After the co-design process, the next step involved training and field-testing the developed program. Care support workers received training on delivering the exercise program, emphasising safety precautions and red flag identification. The training aimed to equip care support workers with the necessary skills to support individuals with dementia effectively. Following the training, the program underwent field testing in home care settings to assess its feasibility and effectiveness in real-world scenarios. Care support workers applied the exercise protocol while working with home care clients and family carers, allowing for the identification of any challenges and gathering feedback for program adjustments. This step focused on the practical implementation of the co-designed exercise program, ensuring its effectiveness in real-life scenarios and addressing any issues identified during the field-testing phase.

The “Safe Functional Home Exercise” program for people with dementia is a 12-week program comprising four exercises: lower limb strengthening exercises, including Exercise A (“knee squats”) and Exercise B (“heel raises”); a balance exercise, Exercise C (“stepping and balance exercises”); and a dual task exercise, Exercise D (“walking and talking exercise”). See Figure 2. Details of the “Safe Functional Home Exercise” program for dementia are provided in the care support worker manual (Appendix A).

The implementation of the exercise program involved the following:Care support workers attended a 5 h training course (hybrid format of in-person attendance and online training) for the exercise program, which was conducted by DCAL, an experienced aged care physiotherapist.Care support workers supervised clients to exercise once weekly, 15 min each week, for 12 weeks during their care shifts.Family carers followed exercise templates (pre-made by the researchers) provided by care support workers and supervised clients to exercise three times a week in addition to the care support worker-supervised session, 30 min each day for 12 weeks, or shared exercise supervision with care support workers to meet the weekly exercise target.DCAL provided fortnightly telephone support to care support workers, family carers and home care clients to ensure exercise progression and safety.All clients commenced their exercise regimen with level 1 exercises, involving hand support, during the first week. Progression to level 2 exercises, which do not require hand support, occurred when they demonstrated the ability to safely accomplish two sets of 8–12 repetitions of level 1 exercises with no more than a 1 min rest between sets.Exercises A, B and C could be made more challenging by reducing hand support or standing base, slowing the movement or adding a hold at the end of the movement, and Exercise D by increasing the difficulty of the talking task (e.g., counting backwards).If the client appeared unsteady or unsafe after progressing to level 2 exercises, the care support workers were instructed to return the client to level 1 exercises. They could increase the difficulty of level 1 exercises if the client was able to. They would review the exercises later to progress the client to level 2.

The resources provided to participants for the exercise program were:Care support workers received (1) an exercise manual (care support worker version); (2) a quick reference guide to the exercise program; (3) a link to an online refresher video of the exercise program (https://youtu.be/xxYQtns9in0); (4) exercise templates that they could provide to family carers for supervising the client to do exercises weekly; (5) an exercise diary to record the exercises they performed with the client; and (6) membership in a WhatsApp or WeChat group for posting questions and sharing their experiences during the program with DCAL, client managers and other care support workers.Clients and family carers received (1) an exercise manual (client and family carer version); (2) an exercise diary to record the exercises they performed with the client; (3) the contact details of DCAL and client managers in case of questions or needing extra support during the exercise program; and (4) a fortnightly telephone support call from the study physiotherapist (trainer) to ensure safety and exercise progression.

Step 3 (Feedback from participants in Step 2) and Step 4 (Refinement of the physical activity program): The original co-design panel that developed the “Safe Functional Home Exercise” program for people with dementia was reconvened after the completion of Step 3. Findings from the surveys (Step 3) and the outcomes of the feasibility study (Step 2) were fed back to panel members to refine and examine different ways to deliver the “Safe Functional Home Exercise” program (Step 4).

### 2.5. Data Analysis

One author, DCAL, translated the surveys that were completed in the Chinese language to the English language before data analysis. Responses to the Likert scale ratings and closed-ended questions were summarised using descriptive statistics. Open-text responses to situations that affected exercise performance, thoughts about adopting the exercise program delivery as a regular practice, and suggestions for enhancing program support, the exercise program and the training course were described using exemplar quotes. Refinements to the exercise program suggested by the co-design panel were summarised.

## 3. Results

Care support workers (n = 19), home care clients (n = 15) and family carers (n = 13) returned the survey. This represented a response rate of 73.1%, 57.7% and 50% for each group, respectively. The demographic characteristics of the participants are shown in Table 1. Among the care support workers, 33.3% were aged in their 40s, 84.2% were female and 68.4% were from a CALD background. The median years (IQR Q_1_, Q_3_) of working in the job role were 2 years (2, 5), and 89.5% completed the 12-week program for some or all of their clients who participated in the program. Among the home care clients, 60% were aged in their 80s, 73.3% were female and 86.7% were from a CALD background. The mean years (SD) for those with a dementia diagnosis were 4 years (2.2), and 93.3% completed the 12-week program. Among the family carers, 61.6% were aged between 50 and 69 years, 61.5% were female and 69.2% were from a CALD background. For those whose care recipients were diagnosed with dementia, the mean duration (SD) since diagnosis was 4.4 years (0.9), and 80% of family carers completed the 12-week program.

The majority of care support workers “strongly agreed” that they felt supported by the research team (89.5%) and their home care organisation (73.7%) throughout the program and that they were overall satisfied with the exercise program (84.2%). Most of the care support workers “strongly agreed” or “somewhat agreed” (ranging from 79% to 90%) with all of the statements about the relevance of the training course and felt they were prepared to prescribe and deliver the exercise program (Figure 3).

All of the home care clients “agreed” that they felt supported by the research team and their home care organisation throughout the program and that they were overall satisfied with the exercise program. The majority of family carers “agreed” that they felt supported by the research team (92.3%) and their home care organisation (76.9%) throughout the program and that they were overall satisfied with the exercise program (76.9%). The majority of home care clients (over 86%) and family carers (over 80%) thought that exercise was important, participation in the exercise program was beneficial for the client and care support workers had received adequate training to deliver the program (Figure 4a,b). However, around 30% of family carers thought that they “were not” given or were “unsure” if they were given sufficient resources and support to help their care recipient do the exercises during the week.

All of the participant groups reported encountering situations that made doing the exercises difficult. Care support workers (31.6%) reported these situations were: “*Covid isolation affecting weekly shift*” (care support worker CSW 5); “*client was distractible*” (CSW 18); “*client’s mood fluctuation*” (CSW 10); “*lacking a visual prompt for Exercise C*” (CSW 17); “*care shift cut and not having allocated time to do the exercises*” (CSW 19); and “*time from training to actually starting with client was too far apart*” (CSW 12). Home care clients (26.7%) reported these situations were: “*medical illness*” (home care client HCC 5); “*attention dividing ex D was difficult*” (HCC 12); and “*mood fluctuations*” (HCC 14). Family carers (41.7%) reported that these situations were: “*client’s temper and not always compliant with exercise instructions*” (family carer FC 3); “*poor memory of the exercises making teaching exercises difficult*” (FC 4); “*client was not cooperative for the exercises*” (FC 9); “*client lacked motivation*” (FC 10); and “*lack of energy, shortness of breath and client’s poor medical condition*” (FC 12).

However, a small proportion of participants reported encountering situations that made doing the exercises easier. Care support workers (27.8%) reported these situations were: “*client’s good mood*” (CSW 5); “*training and support from the trainer (researcher)*” (CSW 12); “*the catch up phone calls made it easier, talking regularly with the researcher and having What’s App to also keep in contact with other support workers*” (CSW 13); and “*doing it with the client and counting out loud (singing the moves)*” (CSW 12). Home care clients (13.3%) reported these situations were: “*When I become more physically agile with the exercises*” (HCC 4) and “*Doing the exercises whilst watching TV*” (HCC 5). Family carers (25%) reported that these situations were: “*the exercises became familiar to us and made us feel we had been trained to progress the exercises*” (FC 6) and “*the exercises were simple and easy to follow that most older people could do*” (FC 8).

Care support workers provided suggestions for improving program support, the exercises and the training course. Most of the comments were complimentary. However, there were a few suggestions for improving the exercises and the training, such as “*…there needs to have funding for support workers to supervise clients to do exercises*” (CSW 8) and “*It will be better if there is a training video developed for client, so that the client can follow the video to do the exercises… and incorporate some game elements*” (CSW10).

Home care clients provided no suggestions for improving program support or the exercises, as all of the comments were complimentary. Family carers provided suggestions for improving program support and the exercises. Most of the comments were complimentary. However, there were a few suggestions for improving program support, such as “*(giving) tips on motivation*” (FC 10) and “*(providing) groups to support carers and home care organisation to take client to a gym or group sessions*” (FC 13), and some suggestions were provided for improving the exercise program, such as “*My mum has difficulty with ex D which involves turning around and has fear of falling*” (FC 3); “*Having a longer exercise program may be better suited for those with less severe dementia e.g., 16 weeks*” (FC 9); and “*Making an online (exercise) program, adding weights for strengthening*” (FC 13).

The majority of care support workers (68.8%) and home care clients (86.7%) thought that the exercises delivered by care support workers during their care shifts should be adopted as a regular practice, compared to 45.5% of family carers who thought the same. Among the participants who responded “no” or “were unsure” to this question, the comments provided by care support workers were: “*The exercise program is very good but I am a bit worried that the client may have accidents during the exercises*” (CSW 6); “*It depends on their level of dementia*” (CSW 17); “*Support workers don’t always have the time allocated to provide the program*” (CSW 18); and “*Because support workers have enough jobs to do in the little time we have with the client. Also we are not qualified physios*” (CSW 19). The comments provided by home care clients were: “*It depends if the trained support worker has the time to continue with this program*” (HCC 2) and “*It depends on the home care package funding if able to support this*” (HCC 9). The majority of the family carers did not provide comments. Among the few comments provided by family carers were: “*It depends on the health condition of the client*” (FC 2); “*It depends on the support worker’s time and funding for this service*” (FC 4); and “*My husband’s condition is poor and may (or may not) need help (with exercises)*” (FC 12).

The original co-design panel was reconvened in the last step (Step 4) of the action research method to refine the exercise program. This panel (n = 10) consisted mostly of the original co-designers from Step 1 (i.e., consumers, physiotherapists and fall prevention researchers), except for new aged care industry stakeholders (home care and clinical governance managers, care support workers) who joined the co-design panel due to staff turnovers since the previous panel. These new panel members had all been involved with the administration and/or delivery of the program in Steps 2 and 3. The co-design panel adopted most of the feedback. As a result, a few minor modifications to the “Safe Functional Home Exercise” program for people with dementia were made (see Appendix B for a summary). These included adding options for different ways to do Exercises A, C and D to make them easier for some clients and making a demonstration video for the exercise program, which could be useful for the participants, especially CALD groups. In addition, there were recommendations for care support workers to take over one or more days of exercise supervision by family carers whenever possible to alleviate the care burden for clients receiving a high-level home care package. The co-design panel did not modify the program to include adding weights for strengthening, providing groups to support family carers and taking clients to gym/group sessions, as they were considered outside the scope of the “Safe Functional Home Exercise” program.

## 4. Discussion

Care support workers, home care clients and family carers who participated in the feasibility study of the “Safe Functional Home Exercise” program for people with dementia rated the exercise program well. Around 80% of care support workers and family carers and 100% of home care clients provided positive feedback on program support and were overall satisfied with delivering or participating in the program. Over 80% of care support workers agreed that the training course was relevant to their knowledge level and learning needs and were confident in prescribing and delivering the exercises to home care clients with dementia. Over 80% of home care clients and family carers thought that participation in the exercise program was beneficial and that the care support workers had received adequate training to deliver the program. There were suggestions for improvement or noted barriers such as making an exercise video for clients, the time availability of care support workers during their care shifts or client-related factors.

The “Safe Functional Home Exercise” program is innovative as it is the first exercise program to our knowledge that has been co-designed for people with dementia. Unlike “off-the-shelf” exercise programs, e.g., the Otago program [21], which are often used by researchers and practitioners for older people who have intact cognition, this program was collaboratively designed for older people with dementia, incorporating insights from both stakeholders in the aged care industry and the consumers themselves. Co-designing a service program has a wide range of benefits for the service’s users and the organisations involved. The benefits can include improving users’ loyalty, reducing costs, increasing people’s wellbeing and organising innovation processes more effectively [22]. This could explain the high degree of acceptability and satisfaction among care support staff, home care clients with dementia or cognitive impairment and their family carers. Indeed, a high degree of adherence to the exercise target was also noted in our feasibility study [16].

Participants reported a number of situations they encountered that influenced their participation in the exercise program. While some situations can occur, e.g., a client’s frustrations/unwillingness to be involved or mood changes, there were some improvements that could be made to the exercise program for future rollouts. For example, the co-design panel recommended adding options for different ways of doing Exercises C and D to suit clients who found the exercise difficult. Future staff training can emphasise different ways that care support workers could use to maximise engagement from clients and improve their motivation to participate in the exercises. An orientation session for family carers was considered to be beneficial in future programs to provide information about the program, strategies to help ensure clients do their exercises weekly and tips for client motivation. Suggestions for adding weights and offering gym/group sessions to home care clients may be a viable option for those who prefer other ways of exercising than participating in the “Safe Functional Home Exercise” program.

Care support workers, home care clients and family carers provided their thoughts on the adoption of program delivery as a regular practice. The perceived barriers to the sustainability of this practice included the time availability of care support workers during their existing home care visits, funding of the home care package to support ongoing exercise and the health conditions of clients, including dementia severity. Leveraging the current workforce of care support workers to enable older people with dementia to engage in physical activities during home care visits holds the potential for scalability and cost-effectiveness in implementation. However, home care organisations planning to introduce this program as a regular practice for clients with dementia would need to examine these factors carefully, and there would be value in incorporating an economic evaluation of a future larger-scale study. Some important implementation aspects for consideration include whether the program would be offered to clients as a 12-week program or a longer program, an intermittent or ongoing program, and which clients and family carers would be most suited to this home-based exercise program. From a policy or organisational perspective, considerations might include determining whether more funding is needed to provide additional time for care support workers, the availability of funding for physiotherapists to train care support workers and provide program support and how the organisation can support families to do this.

There are strengths and limitations of this study that need to be considered. Although we invited all participants of the feasibility study to complete the program evaluation survey, the majority of the surveys were from program completers, which may or may not have resulted in a more favourable evaluation outcome than if all non-completers also returned their surveys. The sample size of the surveys is small due to the sample size used in the feasibility study for each participant group. However, the survey response rate, ranging from 50% to 73%, was regarded as excellent compared to an average online survey response rate of 44% in published research [23]. This was possibly due to a mixture of in-person, mail and online survey delivery methods used to collect the data. Due to funding constraints, we did not conduct interviews with the participants, clients or organisation managers. This would have provided a more in-depth understanding of the acceptability of the program and the sustainability of future programs, especially how this model of care delivery can be aligned with or adapted to a possibly new home care model that is being proposed in Australia.

Looking ahead, the refined program will need to undergo further testing in a larger, definitive study, following the successful application observed in this initial study. The future implementation will be guided by the insights gained from the current research. While the Australian aged care system presently supports various types of home care services through its funding mechanisms, the future translatability of the program will hinge not only on evidence from the larger study and its cost-effectiveness but also on alignment with the prevailing aged care policies at that time.

## 5. Conclusions

The “Safe Functional Home Exercise” program for people with dementia and the associated staff training and resources were well accepted by care support workers, home care clients and family carers. The co-design panel refined the exercise program and suggested options for different ways to deliver the exercises. Future large studies are needed to test the effectiveness and acceptability of the refined program, including an economic evaluation, and to explore ways to scale up the program in individual organisations and more widely in the home care sector.

## Figures and Tables

**Figure 1 healthcare-12-00166-f001:**
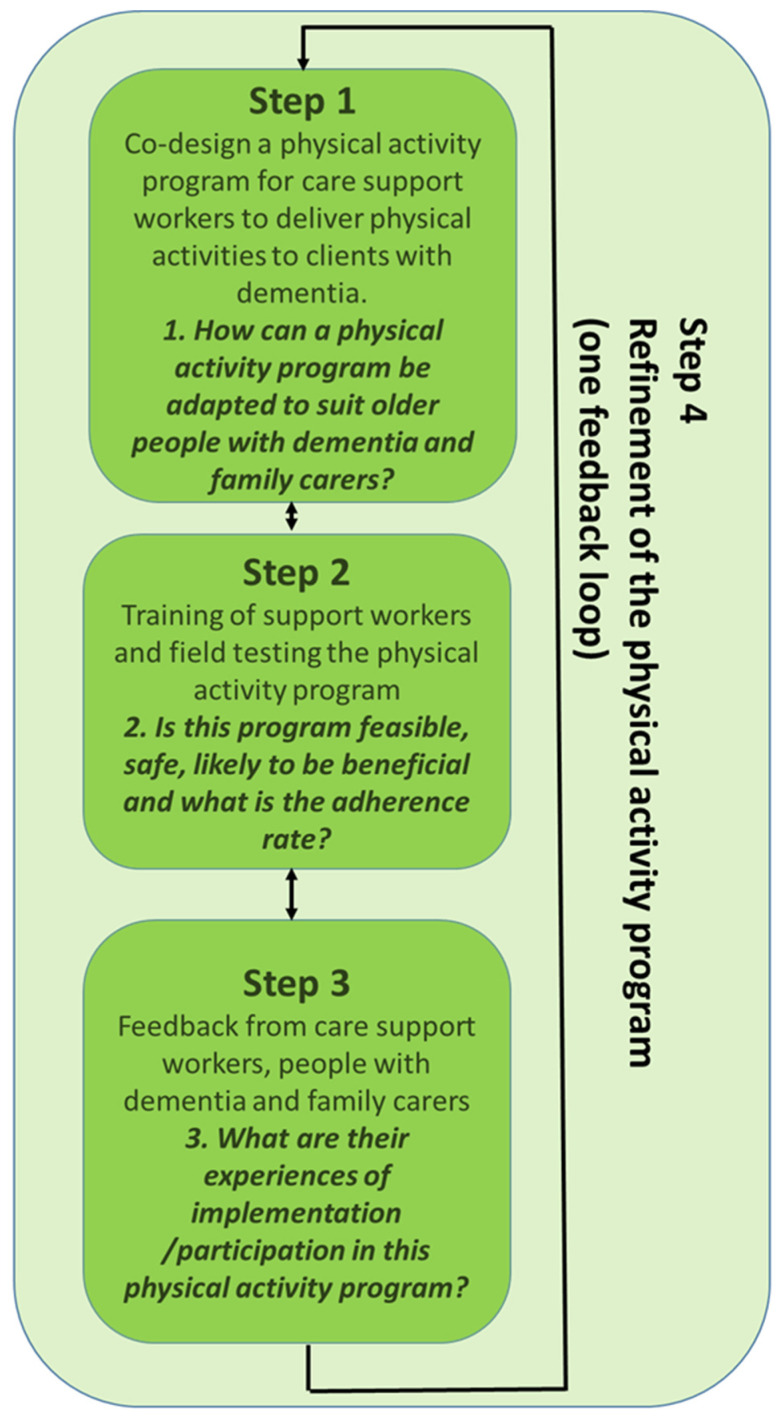
Action research method of the overall study.

**Figure 2 healthcare-12-00166-f002:**
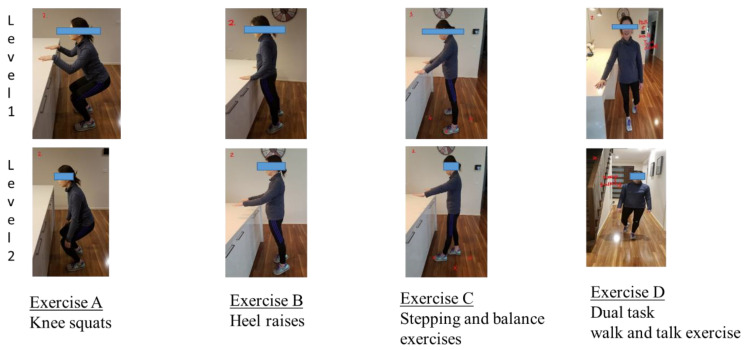
“Safe Functional Home Exercise” program for dementia (level 1 exercises with hand support; level 2 exercises without hand support).

**Figure 3 healthcare-12-00166-f003:**
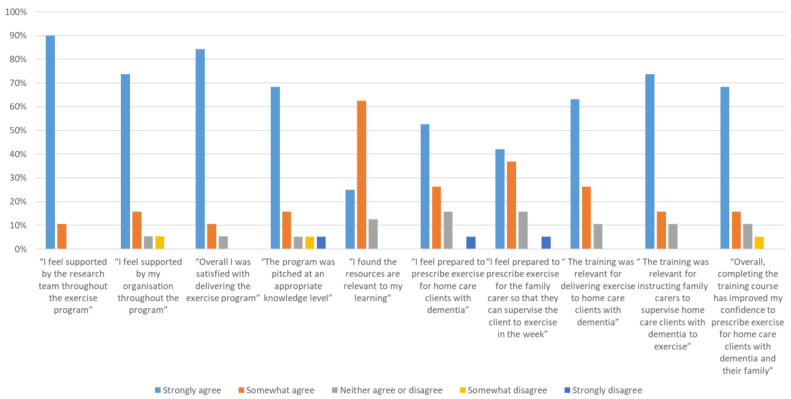
Care support workers’ agreement ratings to program support, program satisfaction, training course and confidence to deliver and prescribe the exercise program.

**Figure 4 healthcare-12-00166-f004:**
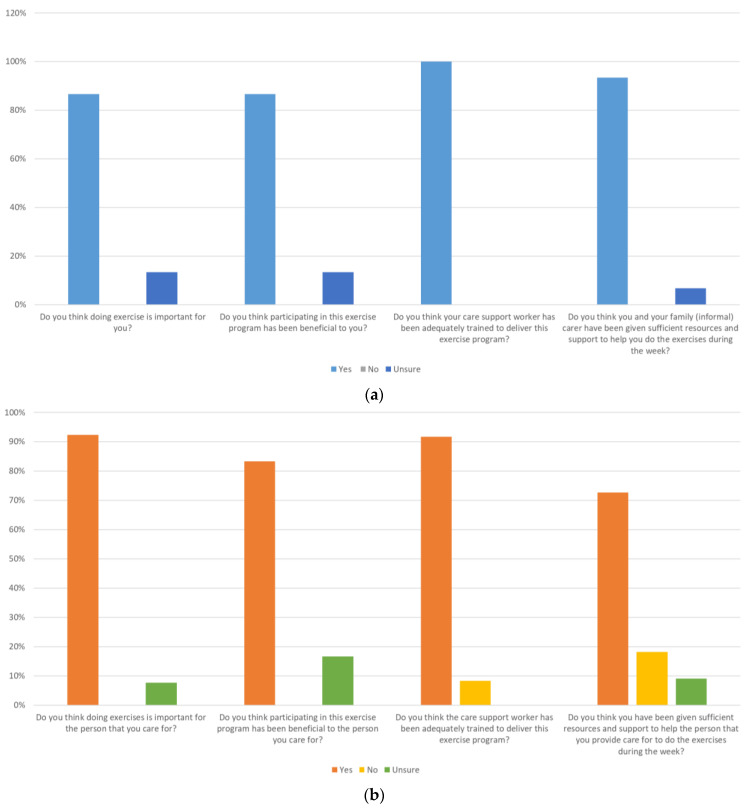
(**a**) Home care clients’ perceptions of the importance of exercise, training of support workers, resources and support to do the exercise program. (**b**) Family carers’ perceptions of the importance of exercise, training of support workers, resources and support to do the exercise program.

**Table 1 healthcare-12-00166-t001:** Demographics of survey participants.

**Care Support Workers**		
	N = 19	N with Data
Age in years—n (%)		18
20–29	0	
30–39	4 (22.2)	
40–49	6 (33.3)	
50–59	4 (22.2)	
60–69	4 (22.2)	
Gender—n (%)		19
Female	16 (84.2)	
Male	3 (15.8)	
Country of birth—n (%)		19
China	12 (63.2)	
Australia	6 (31.6)	
Afghanistan	1 (5.3)	
Language spoken at home—n (%)		17
Chinese (Mandarin/Cantonese/Shanghainese)	12 (70.6)	
English	4 (23.5)	
Dari	1 (5.9)	
CALD ^a^ status—n (%)	13 (68.4)	19
Years lived in Australia—median (IQR Q_1_, Q_3_)	16 (8, 45)	19
Education level—n (%)		19
High school	8 (42.1)	
Trade/Diploma	6 (31.6)	
University and over	5 (26.3)	
Years worked in the care support worker role—median (IQR Q_1_, Q_3_)	2 (2, 5)	17
Completed 12-week program for clients—n (%)		19
Yes, completed program for all	13 (68.4)	
No, completed program for some	4 (21.1)	
No, did not complete for any	2 (10.5)	
Participated in other exercise-training courses for professional development in your role as a care support worker—n (%)		19
No	19 (100)	
Yes	0 (0)	
**Home care clients**		
	N = 15	N with data
Age in years—n (%)		15
70–79	4 (26.7)	
80–89	9 (60.0)	
Over 90	2 (13.3)	
Gender—n (%)		15
Female	11 (73.3)	
Male	4 (26.7)	
Country of birth—n (%)		15
China	13 (86.7)	
Australia	2 (13.3)	
Language spoken at home—n (%)		15
Chinese (Mandarin/Cantonese/Shanghainese)	13 (86.7)	
English	2 (13.3)	
CALD ^a^ status—n (%)	13 (86.7)	15
Years lived in Australia—median (IQR Q_1_, Q_3_)	23 (12, 33)	14
Education level—n (%)		15
Primary school	1 (6.7)	
High school	7 (46.7)	
Trade/Diploma	3 (20.0)	
University and over	4 (26.7)	
Diagnosed with dementia—n (%)		15
Yes	5 (33.3)	
No	8 (53.3)	
Unsure	2 (13.3)	
Years since dementia diagnosis (if diagnosed)—mean (SD)	4 (2.2)	5
Self-reported severity of dementia/cognitive impairment (self-reported)		15
Mild	2 (13.3)	
Moderate	2 (13.3)	
Advanced	2 (13.3)	
I don’t know/was not told	9 (60.0)	
RUDAS score ^b^—Mean (SD)	18.1 (3.7)	15
Completed 12-week program—n (%)		15
Yes	14 (93.3)	
No	1 (6.7)	
Participated in other regular home exercise program in the past		15
No	13 (86.7)	
Yes	2 (13.3)	
**Family carers**		
	N = 13	N with data
Age in years—n (%)		13
50–59	4 (30.8)	
60–69	4 (30.8)	
70–79	1 (7.7)	
80–89	4 (30.8)	
Gender—n (%)		13
Female	8 (61.5)	
Male	5 (38.5)	
Country of birth—n (%)		13
China	9 (69.2)	
Australia	2 (15.4)	
Holland	1 (7.7)	
Sri Lanka	1 (7.7)	
Language spoken at home—n (%)		13
Chinese (Mandarin/Cantonese/Shanghainese)	9 (69.2)	
English	4 (30.8)	
CALD ^a^ status—n (%)	9 (69.2)	13
Years lived in Australia—median (IQR Q_1_, Q_3_)	31 (20, 49)	13
Education level—n (%)		13
Primary school	2 (15.4)	
High school	4 (30.8)	
Trade/Diploma	0 (0)	
University and over	7 (53.9)	
Care recipient diagnosed with dementia—n (%)		13
Yes	6 (46.2)	
No	4 (30.8)	
Unsure	3 (23.1)	
Years since care recipient’s dementia diagnosis (if diagnosed)—mean (SD)	4.4 (0.9)	5
Years caring for the care recipient—median (IQR Q_1_, Q_3_)	6 (4, 10)	13
Self-reported severity of dementia/cognitive impairment of care recipient—n (%)		13
Mild	4 (30.8)	
Moderate	2 (15.4)	
Advanced	2 (15.4)	
I don’t know/was not told	5 (38.5)	
Family carer’s relation with the care recipient—n (%)		13
Spouse/partner	8 (61.5)	
Daughter	3 (23.1)	
Son	2 (15.4)	
Completed 12-week program—n (%)		15
Yes	12 (80.0)	
No	3 (20.0)	
Have you helped the person to do other regular home exercise program in the past?—n (%)		13
No	7 (53.9)	
Yes	6 (46.2)	

^a^ Individuals were categorised as having a CALD (Culturally and Linguistically Diverse) background if they were born outside of Australia and communicated in a language other than English within their household. This classification aligns with the criteria set by the Australian Institute of Health and Welfare (AIHW) in their publication on Culturally and Linguistically Diverse Populations, issued by the Australian Government in 2018. ^b^ Rowland Universal Dementia Assessment Scale (RUDAS). Scores ranged from 0 to 30, with scores 23–30 considered normal, while scores of 22 or less (a lower score indicated greater cognitive impairment) needed to be considered in the clinical context.

## Data Availability

Data are available from the corresponding author upon reasonable request.

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
