# Peer review of "Program Evaluation and Refinement of the “Safe Functional Home Exercise” Program for Improving Physical Activity in Older People with Dementia Who Receive Home Care"

_healthcare, 2024, doi:10.3390/healthcare12020166_

Round 1

Reviewer 1 Report

Comments and Suggestions for Authors

The authors present a manuscript describing a program evaluation of the "Safe Functional Home Exercise" training program for older adults living with dementia. The topic is of public health importance. The program evaluation is described meticulously, the results are depicted clearly with the help of figures, and the results are discussed appropriately. Some minor comments:

1) The study setting is unclear. Steps 1 and 2 have been described in a previous publication cited by the authors. However, it would be easier for the readers to see a high-level summary of these 2 steps in the current manuscript; to provide adequate context to the readers. Was the the co-design panel from the same setting as the survey participants in this manuscript? Why weren't representatives chosen from the current group to participate in the co-design process as stakeholders?

2) Was any specific framework (e.g. the CDC program evaluation framework) used to inform your program evaluation? If so, please describe and cite.

3) Based on the manuscript, it appears that the program was delivered by a research team. Please discuss in the limitations section that the program was led by a team of researchers, ergo its translatability/applicability in real-world, non-research settings remains unknown. Interventions delivered/led by researchers often don't sustain in non-research settings due to a disparate set of challenges. 

Thank you for your contributions.

Comments on the Quality of English Language

n/a

Reviewer 2 Report

Comments and Suggestions for Authors

I believe that it is worthwhile to develop and provide a continuity exercise program for dementia patients receiving home care. However, we are concerned about the following various points, which have led us to this view.

First, let me discuss the definition of the research subjects.

A detailed explanation of "people with dementia" in this study is needed.

For example, the exercise program offered should be different for Alzheimer's disease and Lewy body dementia. Also, the same Alzheimer's disease will have different considerations in the exercise program for early and mid-stage dementia. Please clarify the criteria for the target population in this study.

A detailed description of "trained care support workers " in this study is also needed.

Is this position different from a physical or occupational therapist? Even if one were a physical therapist or occupational therapist, their specialties would be different. Please clarify this point.

"Safe Functional Home Exercise" requires a particularly detailed description.

If you are going to say "safe," then you need to show me all how you are going to keep it safe.

For example, in "Knee squats" of "Exercise A," the flexion angles of the hip and knee joints are large in both Levels 1 and 2. On the other hand, the soles of both feet are fully touching the floor. When holding this posture, the center of gravity tends to shift backward, and backward fall is a concern. In addition, about the "Dual task walk and talk exercise" in "Exercise D," falls during the exercise have been reported even in independent community-dwelling older people. Please describe these concerns so that they can be dismissed. These are the most important points of this study and should be carefully explained.

Number of subjects and statistical analysis

The number of subjects is too small. Therefore, sampling bias cannot be ruled out.

I also believe that the results of the descriptive statistics alone cannot be used to make the claims mentioned in the conclusion.

Without consideration of the above steps, it is difficult to determine the consistency of the authors' views.

That is all.

Please consider revising it.

Round 2

Reviewer 2 Report

Comments and Suggestions for Authors

Thank you for your careful revisions.

I have the impression that it is easier to read than last time.

I would like to add the following two points.

First, I understand that the Rowland Universal Dementia Assessment Scale (RUDAS ) is used to measure the severity of dementia.

On the other hand, representative diagnostic criteria for dementia include the International Classification of Diseases, Tenth Edition ( ICD-10 ) by the World Health Organization, the National Institute on Aging/Alzheimer's Association workgroup ( NIA-AA ) criteria by the U.S. National Institute on Aging/Alzheimer's Disease Association workgroup, and the Diagnostic and Statistical Manual of Mental Disorders, 5th Edition ( DSM-5 ) by the American Psychiatric Association. Association workgroup ( NIA-AA ) and the American Psychiatric Association's Diagnostic and Statistical Manual of Mental Disorders, 5th Edition ( DSM-5 ). For this reason, please explain the rationale for using RUDAS rather than ICD-10, NIA-AA, or DSM-5.

Next, I read your explanation, "A Scalable Program for Improving Physical Activity in Older People with Dementia Including Culturally and Linguistically Diverse (CALD) Groups Who Receive Home Support: A Feasibility Study" doi:10.3390/ijerph20043662. Could you please explain in more detail the differences from this paper in terms of better understanding for many readers?

That is all.

I appreciate your consideration.
